# A Comparison of Deep Neural Networks for Monocular Depth Map Estimation in Natural Environments Flying at Low Altitude

**DOI:** 10.3390/s22249830

**Published:** 2022-12-14

**Authors:** Alexandra Romero-Lugo, Andrea Magadan-Salazar, Jorge Fuentes-Pacheco, Raúl Pinto-Elías

**Affiliations:** 1Tecnológico Nacional de México, CENIDET, Cuernavaca 62490, Morelos, Mexico; 2CONACyT-Centro de Investigación en Ciencias, Instituto de Investigación en Ciencias Básicas y Aplicadas, Universidad Autónoma del Estado de Morelos, Cuernavaca 62209, Morelos, Mexico

**Keywords:** deep learning, monocular depth estimation, unmanned aerial vehicles

## Abstract

Currently, the use of Unmanned Aerial Vehicles (UAVs) in natural and complex environments has been increasing, because they are appropriate and affordable solutions to support different tasks such as rescue, forestry, and agriculture by collecting and analyzing high-resolution monocular images. Autonomous navigation at low altitudes is an important area of research, as it would allow monitoring parts of the crop that are occluded by their foliage or by other plants. This task is difficult due to the large number of obstacles that might be encountered in the drone’s path. The generation of high-quality depth maps is an alternative for providing real-time obstacle detection and collision avoidance for autonomous UAVs. In this paper, we present a comparative analysis of four supervised learning deep neural networks and a combination of two for monocular depth map estimation considering images captured at low altitudes in simulated natural environments. Our results show that the Boosting Monocular network is the best performing in terms of depth map accuracy because of its capability to process the same image at different scales to avoid loss of fine details.

## 1. Introduction

In recent years, the importance of Unmanned Aerial Vehicles (UAVs) has been increasing in sectors such as search and rescue [1], precision agriculture [2], and forestry [3], as they can capture and process different types of data in real time to monitor the environment in which they are moving.

Deep Learning (DL) has proven to be an excellent alternative to achieve autonomous navigation of UAVs, solving a variety of tasks in the areas of sensing, planning, mapping, and control.

One of the main problems that has not yet been solved is giving UAVs the capacity to navigate autonomously at low altitudes in confined and cluttered spaces using cameras as the main sensing sensor.

UAVs with only monocular cameras must be able to detect obstacles to prevent collisions; therefore, it is necessary to create depth maps from RGB images in order to determine the free space to navigate safely. This problem is further complicated when the drone is navigating in a natural environment, due to very thin obstacles such as tree branches or soft obstacles such as large leaves or bushes. Such obstacles can destabilize or damage the UAVs.

Vision-based navigation has been promising for autonomous navigation [4]. First, visual sensors can provide a wealth of information about the environment; second, cameras are very suitable for the perception of the dynamic environment; third, some cameras are cheaper than other types of sensors [5].

The concept of depth estimation refers to the process of preserving the 3D information of the scene using 2D information captured by cameras [6]. A variety of 3D commercial sensors are available to obtain depth information; for example, binocular cameras or LiDAR sensors can obtain accurate depth information, but the memory requirements and processing power are challenges in many onboard applications. In addition, the price of these sensors is very high.

A potential solution to address the problems of 3D sensors could be the use of monocular cameras to create depth maps. The single-view depth estimation technique has currently shown considerable advances in accuracy and speed, increasing the number of approaches published in the literature.

In the specific case of autonomous navigation for drones in outdoor forested environments, Loquercio et al. [6] demonstrated that a neural network for autonomous flight can be trained in simulated environments and perform well in similar real scenarios. However, the creation of such simulated environments can be very time-consuming and there is currently no large set of images with a variety of plants in different settings, such as what may exist in agriculture or forestry.

For this reason, in this work, we evaluated the performance of four models for monocular depth estimation pre-trained on a mixture of diverse datasets with the aim of inferring the depth of RGB images of natural and unstructured scenarios. These models were chosen because they are in a supervised learning framework. We analyzed whether existing databases and network models for depth estimation can generalize to complex natural environments due to the presence of very thin obstacles.

The main contributions of this paper are:A qualitative and quantitative analysis of the performance of four state-of-the-art neural networks for monocular depth estimations of synthetic images of complex forested environments.We propose a new model resulting from the combination of GLPDepth and Boosting Monocular Depth networks, increasing the accuracy of depth maps compared with GLPDepth alone and decreasing the inference time in the Boosting Monocular Depth structure.A discussion of the main open challenges of the monocular depth estimation with neural networks in natural environments.

The rest of this paper is structured as follows. Section 2 introduces the related work. Section 3 explains the network architectures. Section 4 analyzes the comparison of the monocular depth estimation networks. Section 5 enumerates currently open challenges, and in Section 6, we present the conclusions.

## 2. Related Work

During the last decade, DL has produced good results in various areas such as image recognition, medical imaging, and speech recognition. DL belongs to the machine learning computational field and is similar to an Artificial Neural Network (ANN). However, DL is about “deeper” neural networks that provide a hierarchical representation of the data by means of a collection of convolutional layers. This characteristic allows larger learning capabilities and, thus, higher performance and precision [7].

The automatic feature extraction from raw data is the strong advantage of DL, where features from higher levels of the hierarchy are formed by the composition of lower-level features. Thanks to the hierarchical structure and large learning capacity, DL is flexible and adaptable for a wide variety of highly complex challenges [8].

Convolutional Neural Networks (CNN) are a type of architecture that receives input images that are passed through different hidden layers to learn patterns in a hierarchical manner. Therefore, CNN-based models are widely used in the context of collision avoidance [9], especially in the stages related to the perception of the environment and depth estimation [10].

Additionally, with the advance of CNN and publicly available large-scale datasets, the monocular depth estimation method has significantly improved. Current research works have achieved more accurate results with lower computational and energy resources. Accurate depth maps can play an important role in understanding 3D scene geometry, particularly in cost-sensitive applications [11,12].

### 2.1. Monocular Depth Estimation Networks

In recent years, the amount of literature on monocular depth estimation has increased and has evolved to the use of deep learning. Khan et al. [11] and Dong et al. [12] presented two excellent reviews of the state of the art on monocular depth estimation. Deep learning models for monocular depth estimation can be divided by the way they are trained as supervised, unsupervised, and semi-supervised learning models. We focus on supervised learning methods, which are trained using a ground truth to asess their capacity to generalize to other types of environments. Therefore, the latest published papers related to supervised learning deep neural networks were reviewed.

Ranftl et al. [13] trained a robust monocular depth estimation neural network named MiDaS with different datasets obtained from diverse environments. They developed new loss functions that are invariant to the main sources of incompatibility between datasets. They used six different datasets for training and a zero-shot cross-dataset transfer technique to prove generalization. They demonstrated that a model properly trained on a rich and diverse set of images from different sources outperforms the state-of-the-art methods in a variety of settings.

The R-MSFM network [14] is a recurrent multi-scale network with modulation of self-supervised functions, where features are extracted per pixel and a multi-scale feature modulation is performed, iteratively updating an inverse depth via a shared parameter decoder at the fixed resolution. Leveraging multi-scale feature maps prevents error propagation from low to high resolution and maintains representations with modulation semantically richer and more spatially accurate.

The Boosting Monocular Depth network [15] merges estimations in different resolutions with changing context in order to be able to generate depth maps of several megapixels with a high level of detail using a previously trained model. For experimentation, two model options, MiDaS [13] and SGR [16], are taken for depth estimation at high and low resolution; after obtaining both maps, these are combined using a Pix2Pix architecture network [17] with 10 U-net network layers [18] as a generator. Finally, the LeReS model [19] is implemented for depth estimation, which, at the time, has achieved the best performance in the state-of-the-art with AbsRel = 0.09 evaluated with the NYU dataset.

GLPDepth architecture [20] uses both global and local contexts through the entire network to generate depth maps. Their transformer encoder learns the global dependencies and captures multi-scale features, whereas the decoder converts the extracted features into a depth map using a selective feature fusion module, which helps to preserve the structure of the scene and to recover fine details.

On the other hand, Kendall and Gal [21] proposed a Bayesian deep learning framework to estimate depth, in which they modeled two types of uncertainty: aleatoric (concerning noise in observations) and epistemic (related to model parameters), making the networks perform better despite being trained on noisy data. Modeling uncertainty is of great importance in computer vision tasks to know whether the network generates reliable predictions. The generation of confident depth maps is also required for autonomous vehicles in order to make safe and accurate decisions in their environment.

### 2.2. Datasets for Monocular Depth Map Estimation

Research related to autonomous navigation of UAVs has currently focused on indoor, urban, and wooded environments. However, most of the publicly available datasets of the outdoors do not correspond to low-altitude flights. Traditional datasets used to train obstacle detection models [6,22,23,24,25,26,27,28,29,30,31,32,33,34] are KITTI [35] or NYU-v2 [36]. The IDSIA dataset [37] could be considered as a pioneer in the use of images of natural environments, specifically of forest trails.

In addition to these datasets, images obtained from simulators such as Gazebo [38] or AirSim [39] can be used to deal with the lack of data. The TartanAir dataset [40] is one of the best options of data collected from photorealistic simulations of forest environments. A special goal of the TartanAir dataset is that it focuses on the challenging environments with changing light conditions, adverse weather, and dynamic objects. Another alternative is the Mid-Air dataset [41], which is also obtained by a simulated drone flying in natural environments at low altitude.

## 3. Monocular Depth Estimation Networks Analyzed in This Work

The networks that were selected for this analysis are those that have presented the best results in the state-of-the-art to date. In Table 1, we show the AbsRel (Absolute Relative Error) and RMSE (Root-Mean-Square Error) evaluation metrics reported for the networks described above. However, each of the works carried out their experimentation with different datasets, so it is difficult to compare them. Therefore, we perform an evaluation using the different networks and a dataset that contains images of complex natural environments collected by a simulated drone flying at low altitude.

Ranftl et al. [13] demonstrated that a network trained with different datasets can generate a better estimation of depth; therefore, on their experimentation, they started with the experimental configuration of [42] and used a multi-scale architecture based on ResNet for the prediction of depth with a single image. During the experimentation, the influence of the encoder on the architecture was evaluated, so they exchanged different encoders such as ResNet-101, ResNeXt-101, and DenseNet-161. They showed that network performance increased when using higher-capacity encoders. Thus, in subsequent tests, they used ResNeXt-101-WSL, which is a ResNeXt-101 version pre-trained with a massive corpus of Weakly-Supervised Data (WSL). After the evaluation of the encoder, their new model MiDaS was trained with six different datasets (DIW, ETH3D, Sintel, KITTI, NYU, and TUM). The model performed best in the ETH3D dataset according to absolute relative error (Abs Rel).

The MiDaS network has been continuously upgraded to the MiDaS v3.0 DPT version, where it uses the Dense Prediction Transform (DPT) model from the work of [43]. This model can use keys, queries, or values to completely trust the attention ratio between units to find the ratio between each unit in the sequence.

The architecture proposed in [14] consists of four main components: (a) a depth encoder that extracts the pixel representations of ResNet18 except for the last two blocks, which produces multi-scale features with an input resolution of $\frac{1}{2},\frac{1}{4},\frac{1}{8}$; (b) a shared-parameter depth decoder that iteratively updates a zero-initialized inverse depth, preventing spatial inaccuracy at the coarse level from propagating to the fine part; (c) a parameter-learned oversampling module that adaptively oversamples the estimated inverse depth, preserving its motion limits; (d) a Multi-Scale Feature Modulation (MSFM) component that modulates the content in multi-scale feature maps, maintaining semantically richer and spatially more accurate representations for each iterative update. This architecture reduces the number of parameters to 3.8M, making it more suitable for memory-limited scenarios and giving it the ability to process 640x192 videos at 44 frames per second on an RTX2060 GPU.

In [15], an algorithm called double estimation was proposed, where two depth estimations from the same image at different resolutions are merged, obtaining a structure with high-frequency details. Their experiments showed that at low resolutions, depth estimates exhibit a consistent structure of the scene, but high-frequency details are lost. On the other hand, at high resolutions, fine details are well preserved but the scene structure starts to show inconsistencies. To combine the features of both images, they used Pix2Pix4DepthModel, which has a Pix2Pix architecture [17] with U-net layers [18] as a generator and a “PatchGAN” convolutional classifier (only penalizes the scale structure of image patches) as a discriminator. Both the generator and the discriminator use modules of the form Convolution-BatchNorm-ReLU. The entire network process can be described in three steps; first, the network generates a base estimation using the double estimation for the whole image. Then, patch selection starts by tiling the image at the base resolution with a tile size equal to the receptive field size and a $\frac{1}{3}$ overlap; for each patch, a depth estimate is generated using again the double estimation algorithm. Finally, the generated patch-estimates are merged onto the base estimate one by one to generate a more detailed depth map.

In [20], a new architecture that is mainly made up of an encoder, a decoder, and skip connections with feature fusion modules was suggested. The encoder has the objective to take advantage of the rich global information to model long-range dependencies and capture multi-scale context features with a hierarchical transformer [44], where the transformer allows the network to expand the size of the receptive field. The input image is embedded as a sequence of patches with the 3 × 3 convolution operation; then, these patches are used in the transforming block that is made up of several self-attention sets and a Multilayer-Convolution-Multilayer with residual skip. In the lightweight decoder, the channel dimension of the function is reduced to $N_{c}$ with 1 × 1 convolution; then, consecutive bilinear upsampling is used to expand the function to size $H\times W\times N_{c}$. Finally, the output goes through two convolution layers and a sigmoid function to predict the depth map H×W×1, which is multiplied with the maximum depth value to scale in meters. To further exploit local structures in fine detail, a skip connection (to create smaller, receptive fields that help focus on short-distance information) was added with a proposed fusion module.

## 4. Comparison of the Monocular Depth Estimation Networks

The experimentation was performed using the TartanAir public dataset [40], which consists of different trajectories of simulated natural environments captured by a drone. We decided to use the trajectories that are composed of elements of our interest, such as various types of plants (trees, bushes, and grass) and farm tools (fences, lights, poles, and walls).

### 4.1. TartanAir Dataset

The three selected trajectories were: Gascola, Season Forest, and Season Forest Winter. Each of the trajectories is composed of the depth, RGB, and segmentation images obtained by a stereo camera. Table 2 shows the details of each dataset by trajectory.

The main visual characteristics of the three selected environments are described below:

Gascola: A wooded environment with several rocky areas, mossy regions, and areas with different species of pines. Images were captured in daylight in the morning (see row 1 of Table 3).

Season Forest: A wooded environment in the autumn season. Row 2 of Table 3 shows different species of trees with shades of autumn in the trunks and leaves, with effects of falling leaves. The lighting was at noon, so there are shadows under the trees.

Season Forest Winter: A forest area in the winter season; it contains different species of trees showing only the trunks and branches in a greyish shade and covered with snow. Row 3 of Table 3 shows that the capture was made at dusk, so there are many shadows in the area around the trees.

### 4.2. Metrics for Monocular Depth Estimation

In order to evaluate the performance of the networks, we used the following state-of-the-art metrics: Absolute Relative Difference (AbsRel), Root-Mean-Square Error (RMSE), RMSE (log), Square Relative Error (SqRel), and Delta Thresholds ($\delta^{i}$), that is, the percentage of pixels with relative error under a threshold controlled by the constant i [11]. These metrics are defined as follows, respectively:(1)$$AbsRel=\frac{1}{N}\sum^{\;}\frac{\left|d_{i}-d_{i}^{*}\right|}{d_{i}},$$
(2)$$RMSE=\sqrt{\frac{1}{N}\sum^{\;}{\left|d_{i}-d_{i}^{*}\right|}^{2}},$$
(3)$$RMSE\left(\log\right)=\sqrt{\frac{1}{N}\sum^{\;}{\left|logd_{i}-logd_{i}^{*}\right|}^{2}},$$
(4)$$SqRel=\frac{1}{N}\sum^{\;}\frac{{\left|d_{i}-d_{i}^{*}\right|}^{2}}{d_{i}},$$
(5)$$Accuracy\;with\;threshold\;\left(\delta \;<thr\right):\%\;of\;d_{i}\;\;such\;that\;\max\left(\frac{d_{i}}{d_{i}^{*}},\frac{d_{i}^{*}}{d_{i}}\right)<thr,\;where\;thr=1.25,1.25^{2},1.25^{3},$$
where $d_{i}$ and $d_{i}^{*}$ are the ground truth and predicted depth at pixel i, respectively, and N is the total number of pixels. In (5), the accuracy is calculated under a predefined threshold [30,31], where a point of the image $d_{i}$ is used as a positive or negative sample based on how close the ground truth depth of the corresponding pixel is to the depth in the predicted image $d_{i}^{*}$, if their ratio is close to 1. Even though these statistics are good indicators for the general quality of the predicted depth map, they could be elusive. In addition, it is of high relevance that depth discontinuities are precisely located. Therefore, in [16], Xian et al. proposed the ORD metric (Ordinal Relation Error in the depth space) for the evaluation of zero shot crossed datasets. This ordinal error is a general metric for evaluating the ordinal accuracy of a depth map and can be used directly with different sources of the ground truth. The ordinal error can be defined as:(6)$$ORD=\frac{\Sigma_{i}\omega_{i}I\left(\ell_{i}\neq \ell_{i,\tau }^{*}\left(p\right)\right)}{\Sigma_{i}\omega_{i}}$$
where $\omega_{i}$ is set to 1, and the ordinal relationships $\ell_{i}$ and $\ell_{i,\tau }^{*}\left(p\right)$ are computed using Equation (7).
(7)$$\ell =\{\begin{array}{c} +1,\frac{p_{0}^{*}}{p_{1}^{*}}\geq 1+\tau ,\\ -1,\frac{p_{0}^{*}}{p_{1}^{*}}\leq \frac{1}{1+\tau },\\ 0,\;otherwise. \end{array}$$
where τ is a tolerance threshold, and $p_{1}^{*}$ denotes the ground truth pseudo-depth value. When the pair of points are close in the depth space, i.e., $\ell_{i}=0$, the loss encourages the predicted $p_{0}$ and $p_{1}$ to be the same; otherwise, the difference between $p_{0}$ and $p_{1}$ must be large to minimize the loss.

Moreover, Miangoleh et al. [15] proposed a variation in the ordinal ratio error, which they called Depth Discontinuity Disagreement Ratio ($D^{3}R$), to measure the quality of high-frequency depth estimates. Instead of using random points for ordinal comparison as in [16], they used the centers of the superpixels calculated from the depth maps of the ground truth, as well as the centroids neighborhoods, to compare the depth discontinuities. Therefore, this metric focuses on the boundary accuracy to capture performance around high-frequency details.

### 4.3. Qualitative Analysis of Networks

We used some of the RGB images from the three trajectories to estimate their respective depth maps using the pretrained networks: R-MSFM, MiDaS, GLPDepth, and Boosting Monocular Depth (MiDaS). To perform the qualitative analysis, the inferred depth maps were compared with the ground truth of each dataset.

Table 4 illustrates the depth maps that were obtained from the networks using nine test images representative of the different natural obstacles in the environments. The R-MSFM network can define the obstacles that are close and the free space that the scene has in the depth maps. However, in areas of the image where there are several branches with many leaves at different depths, the network considers them as a single object, generating very large areas marked as obstacles.

We used the MiDaS network in its hybrid version with the DPT model because there is a great improvement in the detection of fine details compared with the R-MSFM network. Tree trunks are well defined, but branches with many leaves and distant objects do not always perform well.

In the case of depth maps obtained with the GLPDepth network, the branches and leaves appear thicker than they really are, which could provide a safety margin when navigating between trees. According to the limited color range of the depth maps with this network, we can infer that GLPDepth has some issues determining the depth of the different objects in some of the images of Gascola and Season Forest Winter trajectories.

Finally, with the Boosting Monocular Depth network using the first version of MiDaS, depth maps are obtained with all the obstacles detected; in some scenarios where the branches are in the foreground, they are well defined, reflecting their thickness. This is revealing as it could allow for trajectory planning.

In summary, the above experimental results show that GLPDepth and Boosting Monocular Depth (MiDaS) are the best options to detect obstacles caused by trunks and branches with dense or sparse foliage, regardless of how close or far they are.

Recently, it has been shown that changing the MiDaS network to LeReS in the Boosting architecture improves the performance of depth map generation [45]. Hence, we also decided to carry out an evaluation between GLPDepth and Boosting Monocular Depth with LeReS.

Table 5 displays the depth maps obtained by the two networks. As in the previous test, GLPDepth demonstrates its ability to detect fine details, but these are not very well delimited, thus having in the map the branches and leaves that are thicker than they really appear. Meanwhile, Boosting Monocular Depth (LeReS) shows a great improvement in detecting fine details compared to GLPDepth. The thickness of branches, trunks, and leaves corresponds to what is visualized in the RGB images. In addition, this combination makes a better differentiation of the various depths of the objects.

For the above, we observed that the dataset used in the training phase of each neural network significantly influences the results. The GLPDepth network was trained with the NYU Depth V2 dataset, which contains images of several closed scenarios, and is, therefore, severely affected by light and shadows originating from outdoor environments. Boosting Monocular Depth (MiDaS) was trained to transfer the fine-grained details from the high-resolution input to the low-resolution input using the Middlebury2014 (23 pairs of high-resolution image pairs of interior scenes) and Ibims-1 (high-quality RGB-D images of indoor scenes), whereas Boosting Monocular Depth (LeReS) was trained using various RGBD image datasets, providing depth maps with better definitions of fine details without being affected by shadows or lighting.

### 4.4. Quantitative Analysis of Networks

To measure the performance and inference times of the neural networks, we used a computer with an Intel Core i5-11400H of 2.70 GHz, 16 GB of RAM, a Nvidia GeForce RTX3050 graphics card of 8 GB, and the Windows 10 operating system for all tests.

The results obtained using the three trajectories are shown in Table 6. The GLPDepth network gives the best results in the accuracy metrics, and the Boosting Monocular Depth network shows the best results for the others metrics. The error metrics are better when they tend to 0 and accuracy metrics are better when they are closer to 1. In Season Forest and the Season Forest Winter trajectory, the Boosting Monocular network achieves the best results in the accuracy metric and in the other error evaluation metrics.

We confirm from the observations made in the qualitative analysis that the GLPDepth network is good at detecting obstacles as is shown on the accuracy threshold metrics; they are comparable to those obtained with Boosting Monocular Depth (LeReS). However, this network is not providing a good estimation of depth, because there is a considerable increase in the values obtained with the metrics ORD and $D^{3}R$ in the three trajectories analyzed.

The datasets with which the networks were trained may affect their performance in this forested environment under study. The GLPDepth network was trained with indoor images, while a set of images from a variety of environments were used to train the Boosting network.

The inference times obtained by GLPDepth and Boosting Monocular Depth (LeReS) with the three different trajectories are shown in Table 7. GLPDepth proves to be much faster than Boosting Monocular Depth (LeReS) in all cases. This large time difference is because Boosting has to extract patches from the image and estimate the depth map of each of them, and finally merges them all into a base depth estimation generated from the whole image.

In order to obtain short inference times and to preserve estimation accuracy, we decided to combine the Boosting Monocular Depth network with GLPDepth. We tested this new model only with the Gascola trajectory; the resulting depth maps are shown in Table 8. The new model is better able to highlight fine details in the depth maps than GLPDepth. However, Boosting Monocular Depth using LeReS still generates better depth maps with well-defined fine details than the other networks. Boosting Monocular Depth (GLPDepth) has the same problem as GLPDepth in correctly inferring object depths, so the depth maps also show a very narrow range of colors.

The performance metrics obtained with Boosting Monocular Depth (GLPDepth) are shown in Table 9. This network generates better values in accuracy metrics, and with the other metrics, the Boosting Monocular Depth (LeReS) network achieves the best results.

The inference times obtained using the Gascola trajectory are shown in Table 10. Boosting Monocular Depth (GLPDepth) shows a decrease in inference times compared to Boosting Monocular Depth (LeReS). Nevertheless, these times are still high compared to those generated by GLPDepth.

## 5. Challenges in Monocular Depth Estimation to Navigate in Complex Natural Environments

Many of the depth estimation networks used for inferring depth maps in complex natural environments have been trained using general-purpose datasets, which prevents them from performing well. Thus, it is of great importance to have more freely available datasets of natural environments with images captured at low altitude.

For the UAVs navigation in the free spaces of natural environments, it is necessary to consider the following elements that make navigation difficult: narrow paths, tree trunks, different types of branches and leaves, electricity poles, fences, people, branches in movement, etc.

Depth estimation from a monocular image collected by a flying drone is a complicated problem, as images can have blurring problems, lighting variations, abrupt changes in scale and perspective, and the presence of shadows cast on objects that make up the environment.

Therefore, it is necessary to create models that can create depth maps with high accuracy in spite of the above-mentioned problems and, at the same time, have very short inference times, in order to be embedded in the drone and operate with the limited hardware resources they have. Uncertainty estimation [21,46] is an important aspect to consider when creating depth maps for autonomous navigation to ensure safe flight for the UAV.

## 6. Conclusions

Currently, there has been a breakthrough in the development of deep networks for monocular depth estimation, demonstrating improvements in accuracy and inference times of the models. However, there are still many problems to be solved in the area of low-altitude aerial navigation.

We reviewed four state-of-the-art neural networks for monocular depth estimation networks, with the main idea of using depth maps for obstacle detection for a UAV system in complex natural environments. We found that the networks GLPDepth and Boosting Monocular Depth achieved good performance in detecting fine details in natural environments such as thin branches and small leaves. Nevertheless, Boosting Monocular Depth had high inference times and GLPDepth did not correctly infer the thickness of objects.

We proposed the combination of the GLPDepth network with the Boosting Monocular Depth network for the creation of depth maps. With this new model, inference times were reduced but the accuracy did not improve much compared to Boosting Monocular Depth (LeReS).

We identified some challenges related to the creation of depth maps for autonomous drone navigation in heavily vegetated environments. We also noted the lack of natural environment datasets for training deep neural networks.

## Figures and Tables

**Table 1 sensors-22-09830-t001:** Comparison of monocular depth map estimation networks. The metric values correspond to those reported in their respective publications.

Networks	Datasets	AbsRel	RMSE
MiDaS [13]	ETH3D	0.129	-
R-MSFM [14]	KITTI	0.108	4.470
Boosting [15]	Middlebury	-	0.156
	Ibims-1	-	0.160
GLPDepth [20]	KITTI	0.057	2.297
	Ibims-1	0.200	1.010

**Table 2 sensors-22-09830-t002:** Details of the content of selected TartanAir trajectories.

Details	Gascola	Season Forest	Season Forest Winter
Size of the set	1.38 GB	1.24 GB	1.22 GB
Image resolution	640 × 480	640 × 480	640 × 480
Number of images	382	301	330

**Table 3 sensors-22-09830-t003:** Examples of RGB images from Tartan Air dataset: Gascola (a), Season Forest (b), and Season Forest Winter (c) trajectories.

Trajectories	RGB	RGB	RGB
(a) Gascola	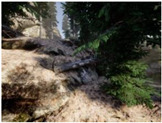	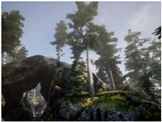	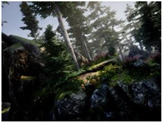
(b) Season Forest	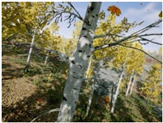	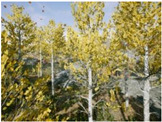	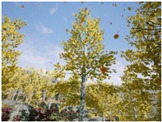
(c) Season Forest Winter	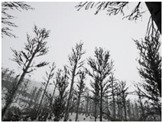	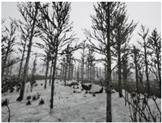	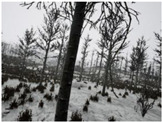

**Table 4 sensors-22-09830-t004:** Depth maps of forested environments generated from state-of-the-art monocular depth networks.

RGB	Ground Truth	R-MSFM	MiDaS	GLPDepth	Boosting Monocular Depth
** 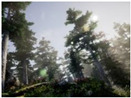 **	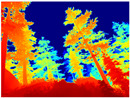	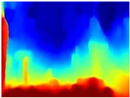	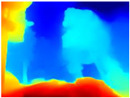	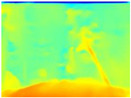	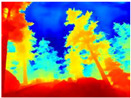
** 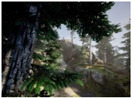 **	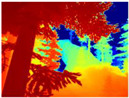	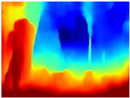	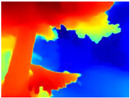	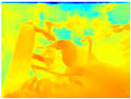	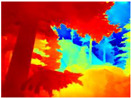
** 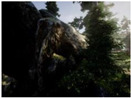 **	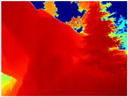	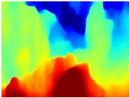	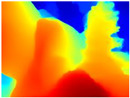	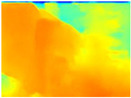	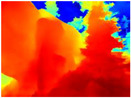
** 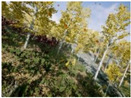 **	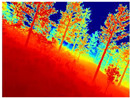	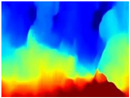	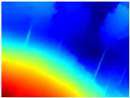	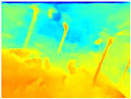	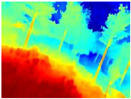
** 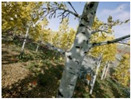 **	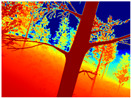	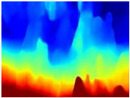	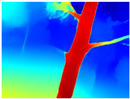	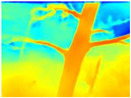	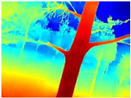
** 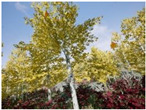 **	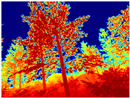	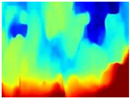	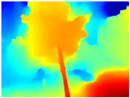	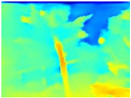	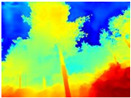
** 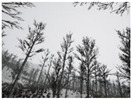 **	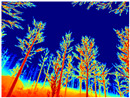	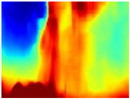	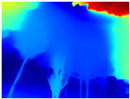	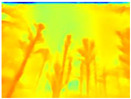	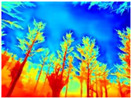
** 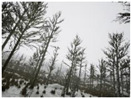 **	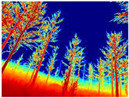	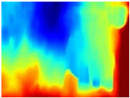	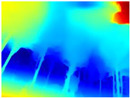	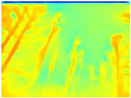	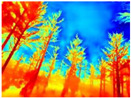
** 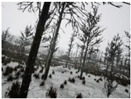 **	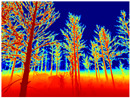	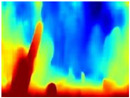	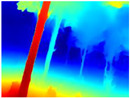	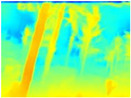	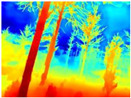

**Table 5 sensors-22-09830-t005:** Depth maps from GLPDepth and Boosting Monocular Depth (LeReS).

RGB	Ground Truth	GLPDepth	Boosting Monocular Depth (LeReS)
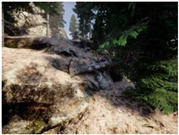	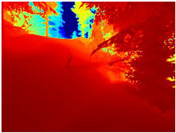	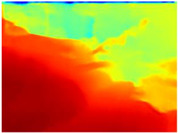	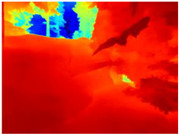
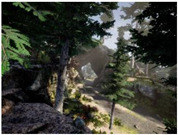	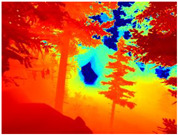	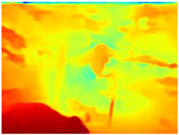	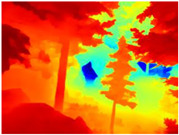
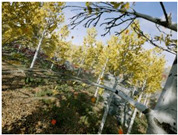	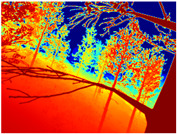	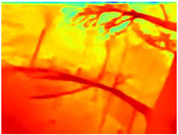	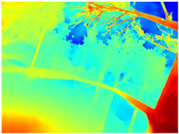
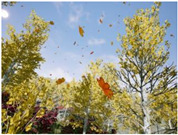	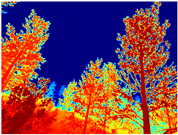	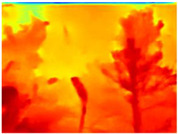	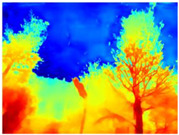
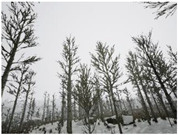	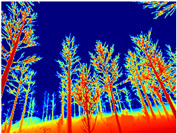	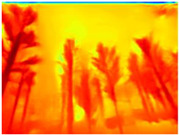	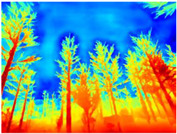
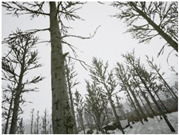	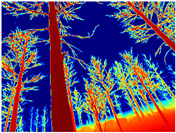	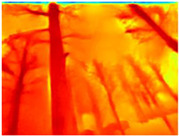	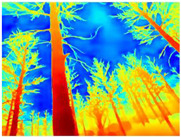

**Table 6 sensors-22-09830-t006:** Comparison results of depth estimation on the three trajectories (Gascola, Season Forest, and Season Forest Winter); the values in bold are the best results using GLPDepth and Boosting Monocular Depth (LeRes).

	Gascola		Season Forest		Season Forest Winter	
Metrics	GLPDepth	Boosting Monocular Depth (LeReS)	GLPDepth	Boosting Monocular Depth (LeReS)	GLPDepth	Boosting Monocular Depth (LeReS)
RMSE	0.086	**0.023**	0.133	**0.089**	0.201	**0.092**
ORD	0.633	**0.295**	0.771	**0.611**	**0.576**	0.624
$D^{3}R$	0.761	**0.489**	0.839	**0.742**	0.695	**0.540**
δ >1.25	**0.771**	0.658	**0.893**	0.908	**0.868**	0.909
δ > $1.25^{2}$	**0.572**	0.392	**0.785**	0.819	**0.736**	0.807
δ > $1.25^{3}$	**0.416**	0.224	**0.673**	0.726	**0.608**	0.696
RMSElog	0.313	**0.205**	0.504	**0.576**	**0.443**	0.507
Abs.Rel.	0.925	**0.487**	2.009	**1.477**	2.165	**1.460**
Sq.Rel.	0.165	**0.073**	0.217	**0.193**	0.255	**0.167**

**Table 7 sensors-22-09830-t007:** Inference runtime comparison on the three trajectories (Gascola, Season Forest, and Season Forest Winter); the values in bold are the best results using GLPDepth and Boosting Monocular Depth (LeRes).

	Gascola		Season Forest		Season Forest Winter	
Runtime (ms)	GLPDepth	Boosting Monocular Depth (LeRes)	GLPDepth	Boosting Monocular Depth (LeRes)	GLPDepth	Boosting Monocular Depth (LeRes)
maximum	**1405**	3424	**796**	2502	**1552**	1851
minimum	**235**	1626	**250**	1578	**124**	1510
σ	215	**167**	163	**122**	230	**42**
average	**514**	1796	**436**	1776	**346**	1604

**Table 8 sensors-22-09830-t008:** Comparison of depth maps generated by GLPDepth, Boosting Monocular Depth (LeReS), and Boosting Monocular Depth (GLPDepth).

RGB	Ground Truth	GLPDepth	Boosting Monocular Depth (LeReS)	Boosting Monocular Depth (GLPDepth)
** 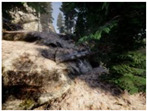 **	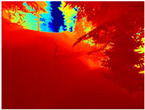	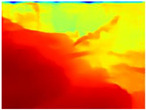	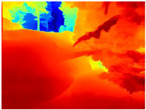	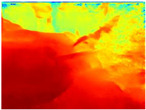
** 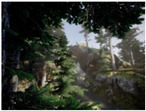 **	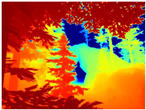	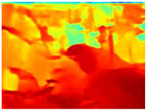	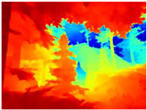	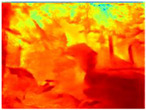
** 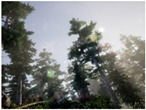 **	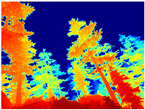	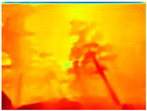	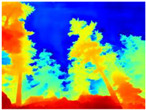	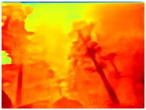
** 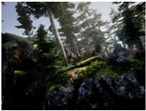 **	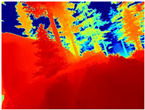	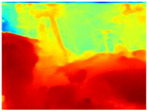	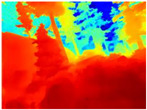	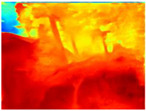
** 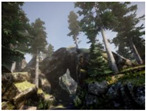 **	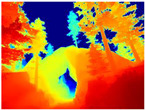	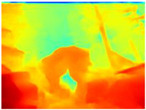	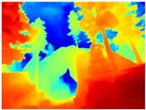	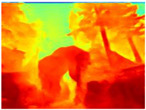
** 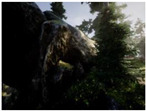 **	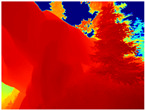	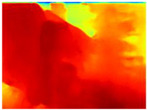	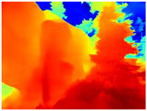	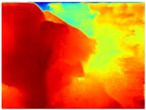

**Table 9 sensors-22-09830-t009:** Comparison results of depth estimation on the Gascola trajectory; the values in bold are the best results using GLPDepth and Boosting Monocular Depth (GLPDepth).

Metrics	GLPDepth	Boosting Monocular Depth (LeReS)	Boosting Monocular Depth (GLPDepth)
RMSE	0.086	**0.023**	0.082
ORD	0.633	**0.295**	0.721
$D^{3}R$	0.761	**0.489**	0.763
δ > 1.25	0.771	0.658	**0.775**
δ > $1.25^{2}$	0.572	0.392	**0.584**
δ > $1.25^{3}$	0.416	0.224	**0.436**
RMSElog	0.313	**0.205**	0.325
Abs.Rel.	0.925	**0.487**	1.280
Sq.Rel.	0.165	**0.073**	0.274

**Table 10 sensors-22-09830-t010:** Inference runtime comparison on Gascola trajectory; the values in bold are the best results using Boosting Monocular Depth with LeReS and GLPDepth.

Runtime (ms)	GLPDepth	Boosting Monocular Depth (LeReS)	Boosting Monocular Depth (GLPDepth)
Maximum	**1405**	3424	2094
Minimum	**235**	1626	1359
σ	**215**	167	086
Average	**514**	1796	1519

## Data Availability

Not applicable.

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
