# Peer review of "A Comparison of Deep Neural Networks for Monocular Depth Map Estimation in Natural Environments Flying at Low Altitude"

_sensors, 2022, doi:10.3390/s22249830_

Round 1
Reviewer 1 Report
Highlight the contribution at the end of Introduction
related work should be improved with more recent work
how equation 5 is obtained
discuss table 6 with more details and table 11
Author Response
We thank the reviewers for their constructive review of our manuscript. We have taken your comments into account and have updated the document. Also, a native English speaker reviewed our manuscript. We provide individual responses to the reviewers below.
We thank the reviewer for the feedback, we have made the following changes:
Introduction: We have updated the description of the contribution in the last paragraphs of the section.
Related work: We have expanded our literature review.
Equation 5: We have clarified the origin of the equation.
Table 6 and 11: We have expanded the description in table 6. We have reassessed the relevance of table 11 to the manuscript and have decided to remove it.
Thank you very much for your time.
Best regards,
The authors.
Reviewer 2 Report
In this paper, the authors address the depth estimation approaches for obstacle detection for low-altitude UAVs.
Is it possible to compare your results with deep reinforcement learning algorithms or LSTM.
Author Response
We thank the reviewers for their constructive review of our manuscript. We have taken your comments into account and have updated the document. Also, a native English speaker reviewed our manuscript. We provide individual responses to the reviewers below.
Thank you for your review. We would like to answer your question:
“Is it possible to compare your results with deep reinforcement learning algorithms or LSTM?”
In our work we consider a recurrent neural network (R-MSFM). In table 4 we show the depth maps generated with this network; however, we did not obtain good results with the analyzed images. Therefore, this network was no longer included in the quantitative analysis.
We also consider only supervised learning methods to evaluate whether existing state-of-the-art databases for monocular depth estimation can generalize to the problem of forest environments with multiple thin objects such as leaves and branches of trees. Consequently, we consider that reinforcement learning techniques are outside the scope of the research. We have made this clear both in the abstract and in introduction.
Thank you very much for your time.
Best regards,
The authors.
Reviewer 3 Report
The authors provide an overview and some empirical studies of four algorithms that are used for monocular depth map estimation. They described the use of 4 algorithms in a comparative study and they conclude that one of them is better in depth map estimation accuracy just because it can process images at different scales than others hence exploiting further detail. The paper has been well organised and presented but am not sure whether that is an article (research) or mini review instead. In addition, I do not think the rationale behind picking out those four algorithms is well articulated - maybe that can be further expanded upon.
Also the same can be said about the dataset used, but the heatmaps are informative indeed.
That's it what it is with this manuscript, given that without a new method presented there is only so much authors can do with this paper. Perhaps try to combine methods, or try a 5th one, etc. In fact it is explicit that their purpose is to run a comprarative study rather than propose an original methodology.
So I think it is what it is with this paper, which is decent in terms of the objectives it has, especially if they incorporate the suggested uncertainty estimation component which is an important attribute to have.
Lastly, given the context of this paper, It might be worth mentioning the potential importance of uncertainty estimation. some indicative references:
-- Kendall, A. and Gal, Y., 2017. What uncertainties do we need in bayesian deep learning for computer vision?. Advances in neural information processing systems, 30.
--De Sousa Ribeiro, F., Calivá, F., Swainson, M., Gudmundsson, K., Leontidis, G. and Kollias, S., 2020. Deep bayesian self-training. Neural Computing and Applications, 32(9), pp.4275-4291.
Author Response
We thank the reviewers for their constructive review of our manuscript. We have taken your comments into account and have updated the document. Also, a native English speaker reviewed our manuscript. We provide individual responses to the reviewers below.
We thank the reviewer for the comments. We have uploaded a revised version. Responses to comments:
“In addition, I do not think the rationale behind picking out those four algorithms is well articulated - maybe that can be further expanded upon.”
In the Introduction section, we have indicated why we have selected the 4 models analyzed and highlighted the contribution of the work. The chosen models have the particularity that their training is based on supervised learning, with the aim of analyzing whether the available public databases allow generalization to our environment of interest.
“That's it what it is with this manuscript, given that without a new method presented there is only so much authors can do with this paper. Perhaps try to combine methods, or try a 5th one, etc. In fact, it is explicit that their purpose is to run a comparative study rather than propose an original methodology.”
We consider only 4 state-of-the-art models, but we propose a combination of 2 of them (GLPDepth and Boosting Monocular Depth). The results of this new model are presented in the last columns of Tables 8, 9, 10. We have modified the wording of the manuscript to make it understandable. We apologize for the confusion caused.
Thank you very much for your time.
Best regards,
The authors.
Reviewer 4 Report
This manuscript presents a comparative analysis of 4 deep neural networks and a combination of two for monocular depth map estimation considering images captured at low altitudes in simulated natural environments. From the evaluation of the networks, the authors detect that the networks GLPDepth and Boosting Monocular Depth have achieved good performance in detecting fine details in natural environments such as thin branches and small leaves. However, the reviewer has some comments.
1. Depth maps from the GLPDepth method in Table 4, Table 5, and Table 8 are very different, the authors should explain them.
2. The reference [29] is not cited.
3. There are also some writing mistakes in the reference part. For example, “Traditional data sets used to train obstacle detection models [7, 13,14,15,16,17,18,19,20,21,22,23,24,25,26,27,28] are KITTI [30] or NYU-v2 [31]. [31]”. Should write as ”Traditional data sets used to train obstacle detection models [7, 13-28] are KITTI [30] or NYU-v2 [31].”
4. The manuscript is difficult to read because it has many typos and grammar errors. The reviewer recommends the authors should check them carefully.
Author Response
We thank the reviewers for their constructive review of our manuscript. We have taken your comments into account and have updated the document. Also, a native English speaker reviewed our manuscript. We provide individual responses to the reviewers below.
Thank you very much for review. We have modified our manuscript to address your comments.
“Depth maps from the GLPDepth method in Table 4, Table 5, and Table 8 are very different, the authors should explain them.”
We have added an explanation of the depth maps obtained by GLPDepth network, highlighting the difference with the other methods.
“The reference [29] is not cited.”
We have added reference number 29.
“There are also some writing mistakes in the reference part. For example, "Traditional data sets used to train obstacle detection models [7, 13,14,15,16,17,18, 19,20,21,22,23,24,25,26,27,28] are KITTI [30] or NYU-v2 [31]. [31]". Should write as "Traditional data sets used to train obstacle detection models [7, 13-28] are KITTI [30] or NYU-v2 [31]."
We have corrected the problem and checked all references.
The manuscript is difficult to read because it has many typos and grammar errors. The reviewer recommends the authors should check them carefully.
The manuscript has been proofread by the authors and revised by a native English speaker.
Thank you very much for your time.
Best regards,
The authors.
Round 2
Reviewer 1 Report
Authors addressed my comments very well
Author Response
Thank you for taking the time to review our manuscript. We have updated the document. A native English speaker has already reviewed our manuscript.
Best regards,
The authors.
Reviewer 3 Report
The Authors did not consider my original suggestion which I have added here verbatim - please make these changes:
'So I think it is what it is with this paper, which is decent in terms of the objectives it has, especially if they incorporate the suggested uncertainty estimation component which is an important attribute to have.
Lastly, given the context of this paper, It might be worth mentioning the potential importance of uncertainty estimation. some indicative references:
-- Kendall, A. and Gal, Y., 2017. What uncertainties do we need in bayesian deep learning for computer vision?. Advances in neural information processing systems, 30.
--De Sousa Ribeiro, F., Calivá, F., Swainson, M., Gudmundsson, K., Leontidis, G. and Kollias, S., 2020. Deep bayesian self-training. Neural Computing and Applications, 32(9), pp.4275-4291.
'
Author Response
Greetings,
We apologize for the omission of your suggestion. We have modified our manuscript to address your comments.
We added the importance of uncertainty estimation in sections 2.1 and 5. We considered the recommended references, which are numbered [21] and [46].
A native English speaker has already reviewed our manuscript and it was updated.
Thanks a lot for you review and the helpful suggestions.
Best regards,
The authors.
Reviewer 4 Report
The authors addressed all of the questions clearly. The revised manuscript is appropriate for Sensors.
Author Response
Thank you for the comments on our manuscript. We have updated the document. A native English speaker has already reviewed our manuscript.
Thanks a lot for your time.
Best regards,
The authors.